# Sodium-Glucose Cotransporter 2 Inhibitors Improve Body Composition by Increasing the Skeletal Muscle Mass/Fat Mass Ratio in Patients with Type 2 Diabetes: A 52-Week Prospective Real-Life Study

**DOI:** 10.3390/nu16223841

**Published:** 2024-11-09

**Authors:** Sara Volpe, Alfredo Vozza, Giuseppe Lisco, Margherita Fanelli, Davide Racaniello, Alessandro Bergamasco, Domenico Triggiani, Giulia Pierangeli, Giovanni De Pergola, Cosimo Tortorella, Antonio Moschetta, Giuseppina Piazzolla

**Affiliations:** 1Interdisciplinary Department of Medicine, School of Medicine, University of Bari “Aldo Moro”, Piazza Giulio Cesare 11, 70124 Bari, Italy; svolpe.doc@gmail.com (S.V.); alfredovozza@live.it (A.V.); margherita.fanelli@uniba.it (M.F.); racaniello.davide@gmail.com (D.R.); alessandro.bergamasco88@gmail.com (A.B.); domenicotriggiani@gmail.com (D.T.); giuliapierangeli@gmail.com (G.P.); cosimo.tortorella@uniba.it (C.T.); antonio.moschetta@uniba.it (A.M.); 2Center of Nutrition for the Research and the Care of Obesity and Metabolic Diseases, National Institute of Gastroenterology IRCCS “Saverio de Bellis”, Castellana Grotte, 70013 Bari, Italy; giovanni.depergola@irccsdebellis.it

**Keywords:** sodium glucose transporter 2 inhibitors (SGLT2is), type 2 diabetes, skeletal muscle mass, body composition, real-life study, sarcopenia, sarcopenic obesity, fat mass, visceral adipose tissue

## Abstract

Background: Sodium-glucose cotransporter 2 inhibitors (SGLT2is) induce body weight loss, but their effect on skeletal muscle mass (SMM) and strength needs to be better elucidated. Objectives: This study aimed to evaluate the effects of SGLT2i on SMM in a real-life population setting of patients with type 2 diabetes (T2D). Secondary outcomes included changes in liver steatosis and in anthropometric and glucometabolic parameters. Methods: Seventy-one patients were treated with SGLT2is as an add-on to metformin for 52 consecutive weeks. Visits were scheduled at baseline (T0) and after 6 (T6) and 12 months of therapy (T12) and included the checking of laboratory tests, measurement of anthropometric parameters, bioimpedance analysis of body composition, and abdominal ultrasound (US). Results: Fat mass (FM) and visceral adipose tissue (VAT) progressively decreased compared to the baseline (FM: −2.9 ± 0.6 kg at T6; −2.8 ± 0.6 kg at T12; VAT: −0.3 ± 0.1 L at T6; −0.4 ± 0.1 L at T12; all *p* < 0.01). Changes in SMM were less pronounced (−0.4 ± 0.3 kg at T6, ns; −0.7 ± 0.4 kg at T12, *p* < 0.05), yielding a beneficial increase in the SMM/FM ratio (+0.3 ± 0.05 at T6 and +0.2 ± 0.05 at T12, all *p* < 0.01). No significant changes in sarcopenia, sarcopenic obesity, fat-free mass, muscle strength, and water compartments were observed at the end of the follow-up period. Anthropometric and glucometabolic parameters, insulin resistance, liver enzymes, and biometric indices and US grading of hepatic steatosis improved throughout this study. Conclusions: In a real-life setting, SGLT2i therapy is associated with weight loss attributable to FM rather than SMM loss without any relevant deterioration in muscle strength. In addition, SGLT2is proved to have beneficial effects on steatotic liver disease.

## 1. Introduction

Sodium-glucose co-transporter 2 inhibitors (SGLT2is), or gliflozins, inhibit the reabsorption of glucose and sodium at the proximal convoluted tubule in the kidney, resulting in iatrogenic glycosuria and natriuresis [1]. Thanks to this mechanism, SGLT2is improve glucose control in an insulin-independent manner and reduce the intraglomerular pressure, glomerular filtration rate, and albumin excretion [2,3]. In addition, SGLT2is reduce blood pressure and uric acid levels, attenuate systemic inflammation due to direct and indirect effects, and provide weight loss, thus reducing the risk of major adverse cardiovascular events (MACEs), the frequency of hospital admission due to heart failure (HF), and both the overall and cardiovascular (CV) mortality [4,5,6].

On the other hand, the chronic loss of calories induced by SGLT2is activate compensatory mechanisms, such as an appetite increase and a rise in glucagon secretion. The consequent activation of lipolysis in the adipose tissue, enhanced free fatty acid flow into the liver, and increased synthesis of ketone bodies can lead to sustained ketosis and the potential onset of normoglycemic ketoacidosis (particularly in the case of beta-cell deficiency or prolonged fasting), a potentially life-threatening condition [7]. SGLT2is can also produce volume depletion and induce glycosuria even in the case of normoglycemia. The mechanism is believed to increase the risk of genital-urinary tract infections especially in patients with associated risk factors, including obesity and uncontrolled T2D, an eventuality that, however, does not usually require SGLT2i discontinuation [8].

Although weight loss is desirable in patients with type 2 diabetes (T2D), who are mostly overweight or obese, healthy weight loss is achieved when a prevalent loss of fat mass (FM) rather than fat-free mass (FFM) is obtained. However, pharmacological treatment is not always supported by specific diet plans and structured training programs in patients with T2D, resulting in an unfavorable influence on skeletal muscle mass (SMM) and strength that plays a crucial metabolic and endocrine role in counteracting hyperglycemia in T2D [9].

As regards to the effects of SGLT2is on SMM, the data are conflicting. A systematic review with a meta-analysis, despite the considerable heterogeneity of the results, highlighted the negative effects of SGLT2is on SMM and skeletal muscle index (SMI), suggesting a possible warning against the use of gliflozins in patients with T2D and sarcopenia [10]. At the same time, neutral or beneficial effects of gliflozins on muscle mass have also been reported [11,12,13,14,15].

The aim of this 52-week real-life study was to evaluate the effect of SGLT2is on body composition, with a specific focus on changes in SMM over the follow-up period, and to compare changes in SSM with changes in FM and visceral adipose tissue (VAT) (SMM/FM and SMM/VAT ratio, respectively) in a population of patients with T2D newly prescribed with gliflozins. The effect of SGLT2is on hepatic steatosis and on anthropometric and glucometabolic parameters was also examined in the same patients cohort.

## 2. Materials and Methods

### 2.1. Study Design, Institution, and Ethics

This 52-week prospective real-life study was carried out at the Metabolic Disorders Outpatients Clinic of the Department of Internal Medicine, University of Bari “Aldo Moro” (Italy), in accordance with the general ethical principles for medical research on humans inspired by the Declaration of Helsinki. The study protocol was formally approved by the Ethics Committee of the University of Bari (n. 6468 version 2_ amendment of 4 August 2022).

### 2.2. Screening for Eligibility of Study Participants

In total, 166 patients with T2D were screened for eligibility from 1 October 2022 to 31 March 2023. The patients were candidates for intensification of antihyperglycemic therapy due to one or more of the following reasons: (a) poor glycemic control, (b) background of cardiovascular and renal risk (secondary CV prevention, or primary CV prevention with high or very high CV risk, HF, renal impairment including reduced glomerular filtration rate, micro- and macroalbuminuria), and excess weight or need of mild-to-moderate weight loss. Among them, 71 patients willing to receive SGLT2is were included in this study: 42 patients were prescribed dapagliflozin (10 mg/die), and 29 empagliflozin (10 mg/die). All patients were on metformin at baseline, at a mean daily dose of 1000 mg. Two patients (both males) discontinued SGLT2is (empagliflozin) by the third month due to genitourinary complaints. A body composition analysis could not be performed in 1 patient at T6 and 2 patients at T12 due to instrumental problems. Furthermore, 2 patients at T6 and 3 patients at T12 did not attend the scheduled visits. The data were analyzed with a treat-to-target approach.

### 2.3. Inclusion Criteria

Inclusion criteria were an established diagnosis of T2D, age > 18 years, stable estimated glomerular filtration rate (eGFR) > 15 mL/min/1.73 m^2^, and eligibility for SGLT2i intensification according to current recommendations and guidelines.

### 2.4. Exclusion Criteria

These included other forms of diabetes mellitus, pregnancy or lactation, inadequate compliance, or contraindications to SGLT2is, an inadequate ability to comply with the follow-up or to provide informed consent, taking oral contraceptives or corticosteroids, viral hepatitis B and C, and excessive ethanol consumption (more than 30 g/day in men and 20 g/day in women); patients who were prescribed SGLT2is or antihyperglycemic medications affecting body composition other than metformin (i.e., glucagon-like peptide-1 receptor agonists (GLP-1RAs)) prior to the study enrollment; implantable electronic devices (cardioverter defibrillators or pacemakers) as indicated by the manufacturer of the segmental multifrequency bioelectrical impedance analysis (SMF-BIA) system we used for the assessment of body composition.

### 2.5. Study Protocol

Eligible patients were fully informed about the study purposes and gave written consent to participate. After the baseline visit (T0), follow-up visits were scheduled semi-annually (T6 and T12). A complete medical history was collected, and a physical examination carried out, at T0 and during each follow-up visit. Clinical and anthropometric parameters included the measurement of office arterial pressure, heart rate, body weight (BW), waist circumference (WC), and body mass index (BMI). Laboratory tests included a complete total blood count and lipid panel, fasting glycemia, glycated hemoglobin (HbA1c), serum creatinine with estimated glomerular filtration rate (eGFR), serum aspartate aminotransferase (AST), alanine aminotransferase (ALT), gamma-glutamyl transferase (γGT), pancreatic isoamylase, lipase and uric acid. Fasting plasma insulin, and C-peptide were also measured. At each time point, the presence of liver steatosis was assessed with a General Electrics Logiq E9 ultrasound (US) machine (GE Healthcare, Milwaukee, WI, USA) and, according to a validated semiquantitative score, was classified as absent (0), mild (1), moderate (2), or severe (3) [16].

Based on laboratory tests and anthropometric parameters, several indices were also calculated. The Homeostasis Model Assessment of Insulin Resistance (HOMA-IR) index, used as an indirect measure of systemic insulin resistance (normal range 0.23–2.5), was calculated as (fasting insulin × fasting glucose)/405.

The Fatty Liver Index (FLI) was used to estimate the presence of liver steatosis, calculated according to the following formula: (e0.953 × loge(triglycerides) + 0.139 × BMI + 0.718 × loge(GGT) + 0.053 × waist circumference − 15.745)/(1 + e0.953 × loge(triglycerides) + 0.139 × BMI + 0.718 × loge(GGT) + 0.053 × waist circumference − 15.745) × 100. An FLI lower than 30 (negative likelihood ratio up to 0.2) suggests that the presence of liver steatosis may be excluded, whereas an FLI equal to or more than 60 (positive likelihood ratio starting from 4.3) was highly indicative of liver steatosis [17].

The Fibrosis-4 index (FIB-4) and the AST-to-platelet ratio index (APRI) were used to estimate the presence of liver fibrosis. The FIB-4 was calculated as follows: FIB-4 = (AST × Age)/(Platelet count × √(ALT)). A cutoff < 1.45 has a negative predictive value of 90% for ruling out extensive fibrosis. A cutoff value > 3.25 has a positive predictive value of 65% for the diagnosis of extensive fibrosis [18]. The APRI score was calculated using the following formula: APRI = (AST/upper limit of normal) × 100/platelet count. A value greater than 0.5 was indicative of an increased risk of liver fibrosis [19].

The study of body composition was performed using a phase-sensitive, octopolar, SMF-BIA (Seca mBCA 525; Seca GmbH & Co., KG, Hamburg, Germany), with frequencies ranging from 1 to 500 kHz, as previously described [20,21,22]. Briefly, the measurements were obtained positioning patients, fasted for 8 h and rested for at least 8 h, in a supine position with legs and arms distanced from the trunk by an angle of 45 and 30°, respectively. The raw data (Rz, Xc) were processed by the Seca Analytics 115 software program to obtain the following values: phase angle (PhA), total body water (TBW), extracellular water (ECW), SMM, skeletal muscle index (SMI) [23], fat mass index (FMI), fat-free mass index (FFMI), VAT.

The hand grip (HG) strength test was performed using a manual hydraulic dynamometer (Lafayette Instrument, Lafayette, IN, USA) to assess the muscle strength in both hands. The best value of three separate measurements, obtained positioning the patient seated with arms flexed at 90, was recorded. The ratio between SMM and HG, namely the muscle quality index (MQI, kg/lg), was used as a functional indicator of skeletal muscle mass.

Finally, patients were considered to have sarcopenia when the SMI value was <6.45 and <8.87 kg/m^2^, with HG strength < 20 and <30 kg, in females and males, respectively [24,25]. To fulfill the criterion of sarcopenic obesity, in addition to being positive for sarcopenia, patients should have a FM% value > 38 in women and >27% in men [26].

### 2.6. Study Outcomes

The primary outcome of this study was to evaluate mean changes in the SMM and muscle mass index (SMI) in patients with T2DM on SGLT2is.

The secondary outcome was to evaluate the impact of SGLT2i on the adipose tissue, in terms of changes in fat mass (FM) and visceral adipose tissue (VAT), with a focus on the SMM/FM and SMM/VAT ratios that are specific components of body composition potentially affecting cardiovascular risk (CVR).

The effects of SGLT2is on anthropometric and glucometabolic parameters, water compartment movements, and fatty liver disease were also analyzed.

### 2.7. Statistical Analysis

To assess whether the real-life available sample was sufficient to obtain reliable results, the first step was to evaluate the power analysis for the variable SMM, which is our primary outcome. With the 71 patients enrolled, as previously specified, and an effect size of 0.6, setting α = 0.05, the power of a repeated measure model was estimated as 0.99 [27].

Patient characteristics at baseline were expressed as mean, standard error, frequency, and percentage, as appropriate. Changes over time (T0, T6 and T12) for each variable were analyzed with repeated-measures mixed models, and changes in the means were estimated by the least-squares method; pre-planned contrasts between times were inserted in the model. To assess the potential effects of age, sex, BMI, HOMA, and type of SGLT2i prescribed, analyses were firstly performed evaluating interaction with the covariates. Associations among categorical variables were evaluated by chi square test. Changes over time in the signs of steatosis and positivity for sarcopenia or sarcopenic obesity criteria were evaluated with the McNemar test. Statistical significance was set to a *p*-value < 0.05.

Statistical analysis was performed with SAS 9.4 software (SAS Institute Inc., Cary, NC, USA), and graphs were produced with Microsoft Excel for Mac (vers.16.16.27, ©Microsoft 2018).

## 3. Results

### 3.1. Baseline Characteristics of Study Population

The baseline characteristics of study participants are shown in Table 1.

Patients were predominantly men (54 of 71; 76%), with a mean age of 68.6 ± 8.4 years. The mean duration of T2D was 9.5 ± 7.9 years, ranging from 0 (new diagnoses) to 36 years. Anthropometric parameters were suggestive of a dysmetabolic state, as confirmed by the values of BMI and waist circumference, with most patients (60.6%) showing a pathological HOMA-IR (≥2.5). Patients with a BMI equal to or more than 30 kg/m^2^, depicting a condition of obesity, accounted for 21% of the study population. The remaining were either normal weight (BMI < 25 kg/m^2^: 24 of 71; 34%) or overweight (BMI 25–29.9 kg/m^2^: 32 of 71; 45%).

Mean values of renal function parameters, lipid profile, liver enzymes, and predictive scores of hepatic fibrosis (APRI and FIB-4) were found to be within the normal range. In contrast, the mean values of FLI, the main predictive score of hepatic steatosis, were above the normal range. Concordantly, a US assessment revealed signs of hepatic steatosis in 80% of patients (grade > 0), of which 52% had mild (grade 1), 34% moderate (grade 2), and 14% severe (grade 3) steatosis (Figure 1).

Finally, the degree of steatosis was associated with the BMI classes (chisq = 10.52 *p* = 0.015). Moreover, an increasing trend toward liver steatosis severity was observed in patients with BMI ≥ 25 kg/m^2^ (Cochran Armitage trend test = 3; *p* = 0.0027).

At baseline, we found that 26.2% of patients had sarcopenia and 16.9% sarcopenic obesity, and both sarcopenia and sarcopenic obesity were not associated with BMI classes. Specifically, 39% of patients with BMI < 25 kg/m^2^ had sarcopenia, a higher percentage than the 19% observed in subjects with BMI ≥ 25 kg/m^2^, although the difference did not reach statistical significance (chi square = 3.1, *p* = 0.08). Moreover, 17.4% of patients in the BMI class < 25 kg/m^2^ vs. 16.7% of those with BMI ≥ 25 kg/m^2^ had sarcopenic obesity according to body composition (chi square = 0.0056, *p* = 0.94).

### 3.2. Changes in Anthropometric, Serologic, and Ultrasonographic Parameters over the Study Period

Changes in anthropometric, serologic, and US parameters during therapy with SGLT2i are listed in Table 1.

As illustrated in Figure 2, treatment with SGLT2i resulted in a significant reduction in the BMI and WC as early as the sixth month of therapy, then confirmed at T12. The mean loss at the end of the follow-up period, as compared with baseline, was −3.5 kg in body weight and −2.6 cm in WC (Figure 2). Patients also showed a decrease in serum uric acid levels, an improvement in glucometabolic control and insulin resistance as expressed by the significant amelioration in fasting plasma glucose, HbA1c, HOMA-IR, triglycerides, total and LDL cholesterol. Improvements were already significant at T6 and then confirmed at T12 (Figure 2).

Changes in eGFR, serum pancreatic enzymes, APRI score, FIB-4, and hand grip strength were negligible throughout the study period (Table 1), while GGT, ALT, and FLI values were significantly lower at T12 than at baseline (Figure 3). In addition, a significant increase in Hb values and hematocrit was found to be evident after 6 and 12 months (see Figure 3).

Consistently with the biohumoral data, the ultrasound evaluation of the liver confirmed a significant beneficial effect of SGLT2is on steatotic disease. Patients were considered ‘improved’ at T12 if grade 0 was maintained or they had a reduction by at least one grade of hepatic steatosis. Accordingly, we found an “improvement” of hepatic steatosis in 51% of patients, a significantly higher percentage than the 20% corresponding to patients who showed no signs of steatosis at T0 (Mc Nemar test = 9.39 *p* = 0.002). This improvement was particularly evident in patients with more significant steatosis at baseline. In fact, the percentage of patients with grade 2 and 3 steatosis decreased up to T12 and none of them had grade 3 steatosis at the end of this study (Figure 4).

Changes over time in all variables were found to be independent of age (≤65 or >65 years), gender (M or F), BMI class (<25 or ≥25 kg/m^2^), HOMA-IR (<2.5, or ≥2.5), and type of SGLT2i prescribed (dapagliflozin or empagliflozin).

### 3.3. Changes in Body Composition over the Study Period

Changes in bioimpedance parameters, describing the body composition during therapy with SGLT2i, are listed in Table 2.

As illustrated in Figure 5, we observed a statistically significant decrease in fat mass and related indices, including the FMI expressed as kg/m^2^ and percentage of FM over the whole body weight, already after 6 months of treatment. These changes were also confirmed at T12. A significant reduction, at both T6 and T12 compared with baseline, was also found with respect to VAT content (Figure 5).

Fat-free mass and its index value (FFMI) remained stable during the study period. However, when expressed as a percentage over the whole body weight, FFMI increased at both times (T6 and T12) compared with baseline (Figure 5). Other relevant components of the FFM, i.e., total body, intracellular, and extracellular body water, were unchanged during the follow-up period with negligible changes in the ECW/TBW ratio over the study period (Table 2 and Figure 5).

Both SMM, expressed in kg, and its index value (SMI, kg/m^2^) were stable until T6, and then showing a slight but significant decrease at T12 (Table 2 and Figure 6), indicating an average decrease in SMM of around 3% compared to the baseline.

Changes in body composition, being more relevant on the FM and VAT than the SMM, consequently induced a beneficial increase in the SMM/FM and SMM/VAT ratios, although the latter change did not reach statistical significance (Figure 6).

No significant changes in either sarcopenia (22.5% at T0 vs. 20% at T12, *p* = 0.7) or sarcopenic obesity (15% at T0 vs. 10% at T12, *p* = 0.41) were observed after one year of SGLT2i therapy in the entire study population. However, changes in these two clinical conditions were also estimated separately in the two BMI classes, with particular attention to the changes occurring in normal-weight patients who had a high incidence of sarcopenia at baseline. We found that in patients with a BMI below 25 kg/m^2^ there was no further deterioration of skeletal muscle mass and strength at T12, as the percentage of sarcopenic patients did not change (43% at T0 vs. 36% at T12, *p* = 0.18) and the percentage of patients with sarcopenic obesity decreased significantly (21% at T0 vs. 7% at T12, *p* = 0.02), compared to the baseline.

## 4. Discussion

The current evidence suggests that abdominal obesity and fat mass excess, especially visceral adipose tissue hypertrophy, are strongly associated with T2D, also contributing to the poor control of glucose control and development of chronic diabetes-related complications [28]. Patients with T2D are at increased risk of sarcopenia, and skeletal muscle deterioration is accelerated compared with age-matched euglycemic individuals [29]. Skeletal muscle deterioration during T2D results in resistance to insulin action on skeletal muscle and impaired glucose uptake, which in turn impair myofibrillar renewal and muscle building, lipid accumulation, mitochondrial dysfunction, and oxidative stress. These effects lead to direct injury to muscle cells, the accumulation of advanced glycation end-products and pro-inflammatory cytokines, intestinal dysbiosis, and hormonal imbalances, in particular related to growth hormone-insulin-like growth factor 1 and the hypothalamic–pituitary–gonadal axes [9]. Overall, patients with T2D are at high risk of sarcopenic obesity, and this risk increases with aging. Specific physical training and nutritional support is therefore recommended [30].

Weight management is an essential goal in the treatment of T2D and should be focused on inducing a decrease in body weight or avoiding unnecessary weight gain. A healthy weight loss is essentially attributable to FM and VAT rather than SMM loss. So far, few studies have examined in patients with T2D the relation between body composition and antihyperglycemic drugs with a proven effect on body weight, such as semaglutide [20,21,22].

In the current study, we examined the effect of SGLT2i treatment on body composition in T2D. It was clinically interesting to verify whether weight loss observed with SGLT2i therapy also resulted in an unfavorable decrease in fat-free mass, particularly muscle mass. SGLT2is induce calorie loss due to glucose excretion in the urine of around 200 kcal/day. The mechanism leads to a sort of starvation and, consequently, activates compensatory mechanisms to restore glucose homeostasis and calorie regain. Apart from an appetite increase, these compensatory mechanisms include the stimulation of glucagon secretion that, in turn, activates hepatic gluconeogenesis, and induces lipolysis in the adipose tissue. In addition, the proteolysis of skeletal muscle myofibrils is activated to supply amino acids as precursors for glucose synthesis through gluconeogenesis in the liver [31]. The latter mechanism can be responsible for muscle mass wasting with potentially detrimental consequences especially in sarcopenic patients [32,33]. To date, the evidence indicates that SGLT2is may reduce SMM during treatment, so caution is usually suggested before prescribing SGLT2is, especially in patients with sarcopenia and physical frailty [34,35,36,37,38,39].

Despite the median BMI value being close to normal values (26.3 kg/m^2^), our patients had a high waist circumference (median 101 cm), indicating a condition of accumulation of visceral adiposity as confirmed by bioimpedance analysis (median VAT values 3.5 L). VAT accumulation is a consistent indicator of overall cardiovascular risk. In addition, the entire cohort showed a high frequency of sarcopenia (26.2%), sarcopenic obesity (16.9%), and hepatic steatosis (80%), indicating an important deterioration of the baseline cardiometabolic status. The decrease in body weight observed during SGLT2i therapy was predominantly attributable to a reduction in fat mass rather than muscle mass, as suggested by the statistically significant increase in the SMM/FM ratio after 6 and 12 months of treatment. Similarly, we observed that the loss of VAT exceeded that of SMM, as demonstrated by the increase in the SMM/VAT ratio after 6 and 12 months of treatment, although this change over time was not statistically significant compared to the baseline. It should be noted that changes in SMM were clinically negligible throughout the study period. In fact, we found a reduction in SMM of about 3% from the baseline to the end of this study, far from the 8% reduction generally considered detrimental to body composition and metabolic balance [40].

Fat-free mass and all the water compartments were also unchanged over the follow-up period, further supporting the non-detrimental effect of SGLT2is on body composition. In line with the present data, previous studies have demonstrated a neutral effect of SGLT2is on muscle mass [11,12,13,14,15,41,42]. A recent randomized trial (EMPA-ELDERLY) conducted in an Asian population aged over 65 years with T2D demonstrated that empagliflozin induced a significant reduction in body weight (−3.26 kg in 52 weeks) without impairing muscle mass and strength in a population at moderate-to-high risk of sarcopenia [43]. In our study, we also found that normal-weight patients (BMI < 25 kg/m^2^), who had a higher incidence of sarcopenia at baseline, did not show further deterioration in body composition with SGLT2i therapy. Indeed, the frequency of patients with BMI < 25 kg/m^2^ and sarcopenia did not change, and that of normal-weight patients with clinical criteria for sarcopenic obesity decreased at the end of the follow-up period. Most importantly, hand grip strength, a parameter that provides functional information about muscle health, did not change significantly during the entire study period.

It is worth noting that we observed a significant increase in the blood concentration of hemoglobin and red blood cells (hematocrit) despite no change in water compartments induced by SGLT2i. Several studies had already demonstrated this increase in hematocrit during the administration of SGLT2is [44,45,46] and associated it with a CV risk attenuation [47], a beneficial effect on renal disease [4], including the prevention of anemia in patients with impaired renal function [48], and on new-onset heart failure [49]. The effect of gliflozins on hematocrit was initially attributed to hemoconcentration as a result of the forced depletion of glucose and sodium in urine and subsequent elimination of water. SGLT2i-induced dehydration is glucose-dependent and occurs soon after the start of treatment, but persists for a few days and then returns to normal values within 92 h [50,51]. Specifically, in the present study, we found that total, intracellular, and extracellular water remained unchanged after up to 52 weeks of consecutive treatment with SGLT2is, and patients showed no signs or symptoms of dehydration during follow-up. Therefore, it is likely that other mechanisms explain the increase in red blood cell concentration. Basic studies demonstrated that SGLT2i can reduce renal medullary perfusion and oxygenation, thereby stimulating renal synthesis and the release of erythropoietin [52]. Similar results emerged from clinical trials employing empagliflozin [53] or dapagliflozin [54,55] showing, with both drugs, increased circulating levels of erythropoietin, improved iron utilization, and reduced biomarkers of inflammation, independently of renal function [48].

Among the secondary outcomes, SGLT2is improved anthropometric parameters (BW, BMI, WC) and glucose and lipid control. We also observed a significant reduction in the serum levels of uric acid, ALT, and GGT without any deterioration of the renal and pancreatic function, as already found in clinical trials and real-life studies [56,57]. Among all the changes we observed with SGLT2is, there was a significant reduction in HOMA-IR by around 40% after one year of therapy compared to the baseline. SGLT2is have been shown to reduce insulin resistance in the skeletal muscle and stimulate angiogenesis, thereby increasing glucose uptake at that site [58] and possibly contributing to the maintenance of skeletal muscle mass and tropism [59]. They were also proven to enhance fat utilization and browning of the adipose tissue, while also attenuating low-grade inflammation and insulin resistance through the activation of type 2 macrophages [60,61], and are considered to improve the efficiency of the respiratory chain by stimulating the mitochondrial utilization of free-fatty acids, which is important for muscle physiology, skeletal muscle, and satellite cell regeneration [62,63,64].

Finally, SGLT2i exerted positive and progressive effects on liver steatosis, as indicated by the reduction over time of FLI, the main biometric surrogate of the disease, which was the only parameter, together with LDL-cholesterol, to further improve at T12 compared to T6. This favorable evidence was also confirmed by the progressive improvement of the US grade of hepatic steatosis, which led, after one year of therapy, to the disappearance of signs of severe disease in patients with grade 3 steatosis at baseline. This finding is particularly important because a high percentage of patients with T2D and steatosis are at risk of or have moderate-to-severe hepatic steatosis, fibrosis, cirrhosis, hepatocellular carcinoma, and liver failure [65]. Consistently with our data, it has been reported that the absolute percentage of hepatic fat content based on magnetic nuclear resonance or controlled attenuation parameter (CAP) improves under SGLT2i therapy, with a trend towards reductions in fibrosis markers, suggesting that SGLT2is might delay the progression of Metabolic Dysfunction-Associated Steatotic Liver Disease (MASLD) [66,67,68,69]. In addition, SGLT2is, with or without metformin, were demonstrated to improve the FIB-4 (from 1.79 ± 1.10 to 1.56 ± 0.75) [70], liver stiffness and steatosis, while also reducing the risk of hepatocellular carcinoma and non-liver cancer, in patients with T2D and MASLD [66,71]. The exact mechanisms by which SGLT2is work on MASLD are currently under investigation. The insulin-sensitizing and anti-inflammatory effects of this class of drugs, as well as the improvement of free-fatty acid utilization within the mitochondrial β-oxidation process, have been thought to be involved in this effect [72]. Furthermore, it has recently been suggested that SGLT2is downregulate the expression of miR-34a-5p and inhibit the expression of genes related to the Transforming Growth Factor (TGF)-beta signaling pathway in hepatic stellate cells, hence negatively affecting the progression of steatosis-associated fibrosis [73].

The strengths of this study are the well-standardized and comprehensive evaluation, long-term follow-up period (52 weeks), and real-life setting. Patients were managed according to well-established procedures, current guidelines, and general recommendations of good clinical practice. Moreover, all patients underwent extensive examinations that included laboratory tests, analysis of body composition, and liver ultrasound. Although patients were not unselected due to the real-life nature of this study, they were willing to undergo intensification with SGLT2is and were included consecutively in the protocol after giving signed written informed consent.

The main limitations are the lack of a control group, the single-center design and the low number of women enrolled, which did not allow a more in-depth analysis of the muscle mass stratified by the gender of the patients. Finally, more accurate procedures to examine hepatic steatosis, stiffness, and fibrosis would have increased the level of evidence for SGLT2is on MASLD (secondary outcome).

## 5. Conclusions

The results of the present study indicate that SGLT2is are effective and safe in patients with T2D, regardless of baseline body weight and body composition. The use of this class of anti-hyperglycemic agents did not affect muscle mass and strength in a clinically significant manner, as the average loss of SMM after one year of therapy was only 3%. Overall, the weight loss induced by SGLT2is was significant but “healthy” because it was mainly attributable to a decrease in fat mass and visceral adipose tissue with improvement in SMM/FM ratio. Moreover, SGLT2is improved liver steatosis, thereby appearing as promising therapeutic tools in the treatment of MASLD in patients with T2D.

## Figures and Tables

**Figure 1 nutrients-16-03841-f001:**
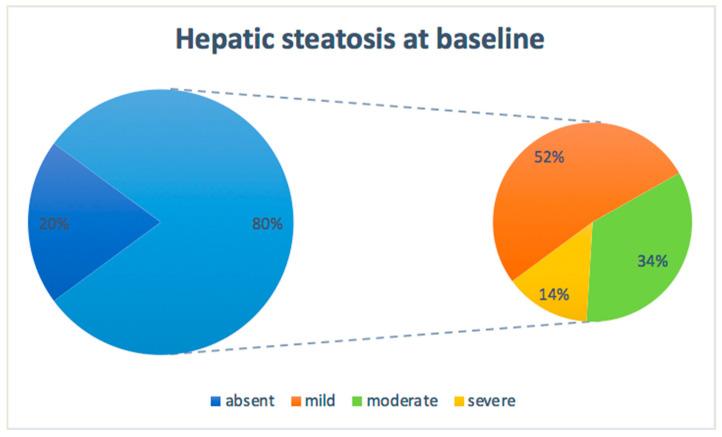
Percentage of patients with different degrees of steatosis at baseline.

**Figure 2 nutrients-16-03841-f002:**
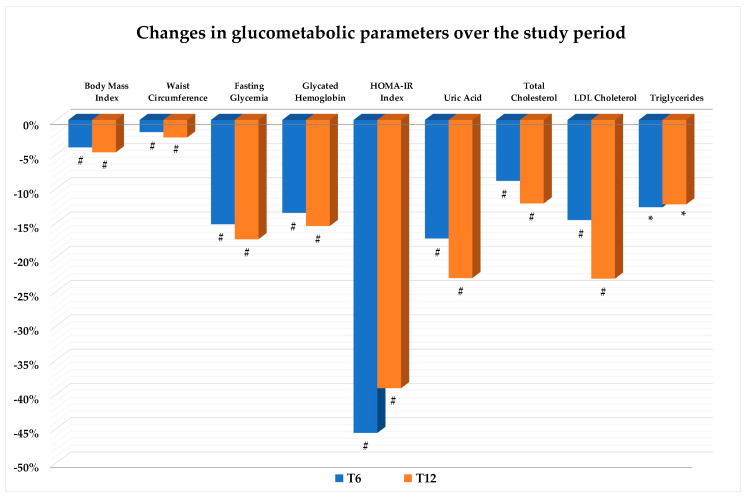
Values are expressed as percent change from baseline (T0). Change vs. T0: * *p* < 0.05; # *p* < 0.01. Abbreviations: HOMA-IR: Homeostasis Model Assessment of Insulin Resistance; LDL: low-density lipoprotein.

**Figure 3 nutrients-16-03841-f003:**
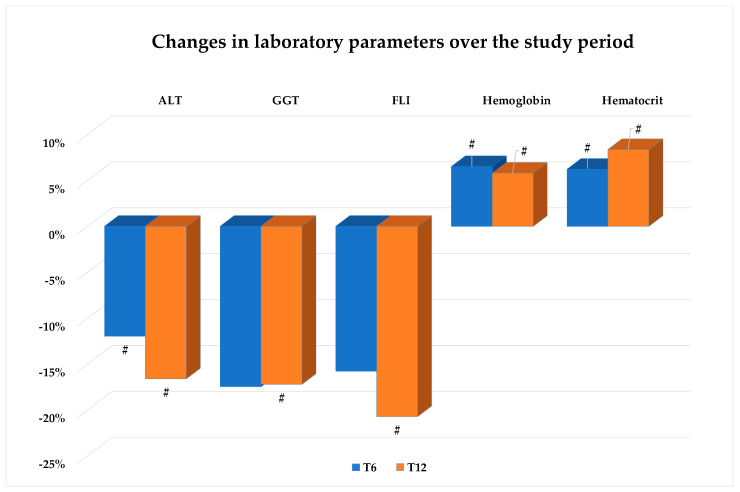
Values are expressed as percent change from baseline (T0). Change vs. T0: # *p* < 0.01. Abbreviations: ALT: alanine aminotransferase; GGT: gamma-glutamyl transferase; FLI: Fatty Liver Index.

**Figure 4 nutrients-16-03841-f004:**
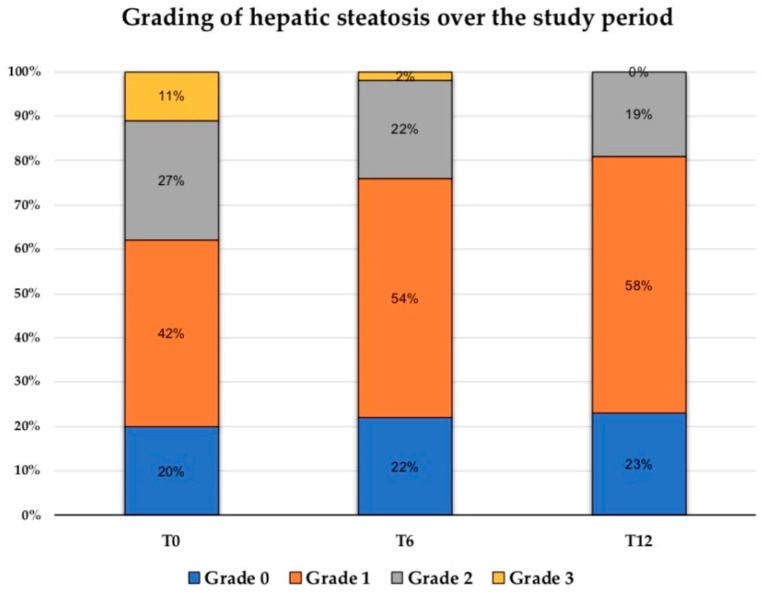
Values are expressed as percentage of patients with different degrees of steatosis over the study period.

**Figure 5 nutrients-16-03841-f005:**
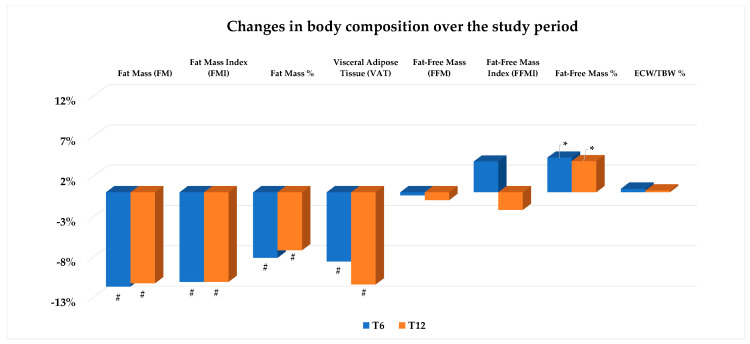
Values are expressed as percent change from baseline (T0). Change vs. T0: * *p* < 0.05; # *p* < 0.01. Abbreviations: ECW: extracellular water; TBW: total body water.

**Figure 6 nutrients-16-03841-f006:**
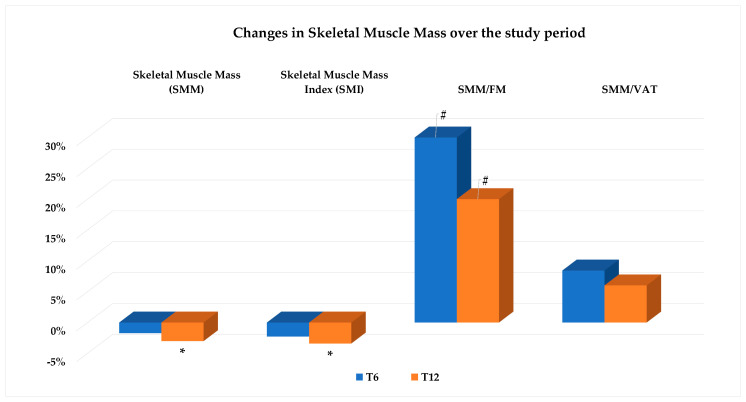
Values are expressed as percent change from baseline (T0). Change vs. T0: * *p* < 0.05; # *p* < 0.01. Abbreviations: FM: fat mass; VAT: visceral adipose tissue.

**Table 1 nutrients-16-03841-t001:** Baseline characteristic of study participants (T0) and estimated means of anthropometric, serologic, and ultrasonographic parameters over the study period (T6, T12).

Parameters	Time		
	T0	T6	T12
Body mass index (kg/m^2^)	27.5 ± 0.5	26.4 ± 0.5 #	26.3 ± 0.5 #
Waist circumference (cm)	101.6 ± 1.3	99.9 ± 1.3 #	99.1 ± 1.4 #
Fasting glycemia (mg/dL)	141.9 ± 6.8	120.4 ± 3.8 #	117.3 ± 2.7 #
Glycated hemoglobin (mmol/mol)	57.5 ± 2.4	49.7 ± 1.3 #	48.6 ± 1.1 #
Fasting serum C-peptide (ng/mL)	3.0 ± 0.2	2.7 ± 0.2 *	2.7 ± 0.2 #
Fasting serum insulin (mUI/L)	12.6 ± 1.2	8.4 ± 0.8 #	9.3 ± 0.8 *
HOMA-IR index	4.6 ± 0.6	2.4 ± 0.2 #	2.6 ± 0.3 #
Uric acid (mg/dL)	5.2 ± 0.2	4.2 ± 0.2 #	4 ± 0.2 #
Total cholesterol (mg/dL)	148.5 ± 4.3	135.2 ± 3.3 #	130.2 ± 3.1 #
LDL cholesterol (mg/dL)	76 ± 3.5	64.5 ± 2.7 #	57.9 ± 2 #, &
HDL cholesterol (mg/dL)	51 ± 2.0	52.3 ± 2.1	52.8 ± 1.9
Triglycerides (mg/dL)	120.1 ± 7.6	105 ± 5.6 *	105.5 ± 5.6 *
Hemoglobin (g/dL)	13.8 ± 0.2	14.7 ± 0.2 #	14.6 ± 0.2 #
Hematocrit (%)	41.8 ± 0.5	44.3 ± 0.5 #	45.2 ± 0.7 #
Serum creatinine (mg/dL)	0.92 ± 0.03	0.94 ± 0.03	0.89 ± 0.03
Glomerular filtration rate (mL/min/1.73 m^2^)	79.8 ± 2.2	79.5 ± 2.3	81.3 ± 2.3
AST (IU/L)	23.7 ± 1.2	20.8 ± 0.9 *	21.7 ± 0.9
ALT (IU/L)	30.1 ± 1.8	26.4 ± 1.7	25 ± 1.7 #
GGT(IU/L)	39.5 ± 4.9	28.1 ± 2.2	28.2 ± 4 #
Pancreatic isoamylase (IU/L)	42.4 ± 4.3	44.7 ± 4.4	50.7 ± 5.0
Lipase (IU/L)	230.3 ± 18.2	222.4 ± 19.2	215.3 ± 18.5
FLI	54.5 ± 3.3	46.3 ± 3.3 #	43.6 ± 3.5 #, &
APRI	0.29 ± 0.01	0.26 ± 0.01	0.27 ± 0.02
FIB-4	1.38 ± 0.05	1.36 ± 0.07	1.45 ± 0.08
Hand grip strength (kg)	31.2 ± 1	30.8 ± 1.1	30.9 ± 1

For each parameter, estimated means and SE at T0, T6, and T12 obtained by mixed-model analysis are reported. Abbreviations: HOMA-IR: Homeostatic Model Assessment for Insulin Resistance; LDL: low-density lipoprotein; HDL: high-density lipoprotein; AST: aspartate aminotransferase; ALT: alanine aminotransferase; GGT: gamma-glutamyl transferase; FLI: Fatty Liver Index; APRI: AST-to-platelet ratio index; FIB-4: Fibrosis-4 index. Change vs. T0: * *p* < 0.05, # *p* < 0.01. Change vs. T6: & *p* < 0.05.

**Table 2 nutrients-16-03841-t002:** Estimated means of bioimpedance parameters at baseline (T0) and over the study period (T6, T12).

Parameters	Time		
	T0	T6	T12
Visceral Adipose Tissue (VAT; L)	3.5 ± 0.2	3.2 ± 0.2 #	3.1 ± 0.2 #
Fat Mass (FM; kg)	24.8 ± 1.1	21.7 ± 1.2 #	21.8 ± 1.2 #
Fat Mass Index (FMI; kg/m^2^)	9 ± 0.4	7.9 ± 0.4 #	7.9 ± 0.4 #
Fat Mass (FM; %)	32 ± 1	29.3 ± 1.2 #	29.5 ± 1.1 #
Fat-Free Mass (FFM; kg)	51.5 ± 1.1	51.3 ± 1.2	51 ± 1.2
Fat-Free Mass Index (FFMI; kg/m^2^)	18.3 ± 0.3	19 ± 0.7	18.7 ± 0.7
Fat-Free Mass (FFM; %)	67.3 ± 1.1	70.3 ± 1.4 *	70 ± 1.3 *
Skeletal Muscle Mass (SMM; kg)	23.2 ± 0.6	22.9 ± 0.7	22.5 ± 0.7 *
Skeletal Muscle Mass Index (SMI; kg/m^2^)	8.8 ± 0.4	8.6 ± 0.4	8.5 ± 0.4 *
SMM/VAT (kg/L)	8.3 ± 0.7	9 ± 0.7	8.8 ± 0.6
SMM/FM (kg)	1.0 ± 0.1	1.28 ± 0.09 #	1.18 ± 0.07 #
Total Body Water (TBW; L)	38.2 ± 0.8	38.1 ± 0.8	37.9 ± 0.9
Extracellular Body Water (ECW; L)	17.4 ± 0.3	17.4 ± 0.3	17.3 ± 0.4
ECW/TBW (%)	45.7 ± 0.3	45.9 ± 0.3	45.9 ± 0.3
Phase Angle (◦)	5.2 ± 0.1	5 ± 0.1	4.9 ± 0.1 *
Muscle Quality Index (MQI) (kg/kg)	1.3 ± 0.04	1.34 ± 0.04	1.37 ± 0.04

For each parameter, estimated means and SE at T0, T6, and T12 obtained by mixed-model analysis are reported. Variation vs. T0: * *p* < 0.05, # *p* < 0.01.

## Data Availability

Data supporting the results are available on reasonable request to the corresponding authors.

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
