# Peer review of "Sodium-Glucose Cotransporter 2 Inhibitors Improve Body Composition by Increasing the Skeletal Muscle Mass/Fat Mass Ratio in Patients with Type 2 Diabetes: A 52-Week Prospective Real-Life Study"

_nutrients, 2024, doi:10.3390/nu16223841_

Round 1
Reviewer 1 Report
Comments and Suggestions for Authors
Review of “Effect of Sodium-Glucose Cotransporter Inhibitors on Skeletal Muscle Mass in Patients with Type 2 Diabetes: A 52-Week Prospective Real-Life Study” (nutrients-3279260)
This study investigated the effect of SGLT2i on the change of SMM in patients with type 2 diabetes and showed that one year of SGLT2i use resulted in less pronounced changes in SMM and increases in SMM/FM ratio. This study was potentially interesting; however, several problems need to be solved.
1. In this study, 71 patients newly used SGLT2i. Were there any dropouts or reported side effects?
2. Table 2 and Table 3. For T6 and T12, not only the change values but also the actual measured values should be noted.
3. Figure 4 shows the change from grade 0~3, but it is not clear which square is which grade.
4. Line 292-300. Mc Nemar T0 vs T12: 0.14, p=0.7, Mc Nemar T0 vs T12: 0.67, p=0.41, Mc Nemar: 1.8, p=0.18 and Mc Nemar T0 vs T12: 5.44, p=0.02. This reviewer doesn't understand what the numbers 0.14, 0.67, 1.8 and 5.44 mean. To show the actual number of sarcopenia or sarcopenic obesity should be added.
5. Furthermore, to show the effect of SGLT2i on the change of each parameter in sarcopenia patients is important information. Thus, sub-analysis with and without sarcopenia should be added.
Author Response
REVIEWER 1
This study investigated the effect of SGLT2i on the change of SMM in patients with type 2 diabetes and showed that one year of SGLT2i use resulted in less pronounced changes in SMM and increases in SMM/FM ratio. This study was potentially interesting; however, several problems need to be solved.
- In this study, 71 patients newly used SGLT2i. Were there any dropouts or reported side effects?
We thank the reviewer for this comment which allows us to clarify that 2 of the 71 patients (both males) abandoned treatment with SGLT2i (empagliflozin) due to genitourinary complaints. They dropped out after 2 and 3 months of therapy, respectively. Moreover, we could not perform body composition analysis (BIA) in 1 patient at T6 and 2 patients at T12 for instrumental problems. Furthermore, 2 patients at T6 and 3 patients at T12 did not attend the scheduled visits. Missing data were handled with an intention-to-treat approach. All this information has been specified in the paragraph 2.2 of the current version of the manuscript (lines 93-98).
Note that the reduction in the number of patients did not affect the power, which remains above 90%, and that changes over time (T0, T6 and T12) for each variable were analyzed, in the current as in the previous version of the manuscript, with repeated-measures Mixed Models, which perform better than other methods in the presence of missing values.
- Table 2 and Table 3. For T6 and T12, not only the change values but also the actual measured values should be noted.
Thanks for the suggestion. Tables 2 and 3 have been renumbered as Table 1 and 2, because Table 1 of the previous version has been removed according to the second reviewer’s suggestion. Tables were modified as requested, showing variable means at T0, T6 and T12. The titles and legends have also been changed accordingly. The changes versus T0, expressed as a percentage, are illustrated in Figures 2, 3, 5 and 6.
- Figure 4 shows the change from grade 0~3, but it is not clear which square is which grade.
We are sorry for the inconvenience. Figure 4 is probably only displayed correctly if opened with an updated version of Adobe Acrobat Reader. Figure 4 shows the following four colors: blue lower band: grade 0 steatosis; orange (band II from the bottom): grade 1; grey (band III from the bottom): grade 2; yellow (band on the top): grade 3. Attached you find the jpeg format of Figure 4.
- Line 292-300. Mc Nemar T0 vs T12: 0.14, p=0.7, Mc Nemar T0 vs T12: 0.67, p=0.41, Mc Nemar: 1.8, p=0.18 and Mc Nemar T0 vs T12: 5.44, p=0.02. This reviewer doesn't understand what the numbers 0.14, 0.67, 1.8 and 5.44 mean. To show the actual number of sarcopenia or sarcopenic obesity should be added.
In accordance with the reviewer's request, in the revised version of the manuscript (lines 304-312) we have only reported the percentages of patients positive for the criteria of sarcopenia or sarcopenic obesity and the statistical significance of the Mc Nemar test. It should be noted that this test only considers patients present at all assessment points, neglecting those present at only one point.
- Furthermore, to show the effect of SGLT2i on the change of each parameter in sarcopenia patients is important information. Thus, sub-analysis with and without sarcopenia should be added.
The reviewer's suggestion is correct, and indeed this is an issue we are already investigating on a larger sample size. The limited number of sarcopenic patients in our study population makes it preferable to perform this sub-analysis on a larger population in order to draw solid conclusions.
Reviewer 2 Report
Comments and Suggestions for Authors
I congratulate you on this interesting and complex study. I would like you to clarify a few points:
1. The title does not reflect the complexity of the studies. Please rephrase
2. In the introduction, the advantages and disadvantages of this class of anti-diabetics should be better pointed out
3. Please specify from the patients initially taken into the study, how many remained in the study at T6 and T12?
4. The statistical methods used to obtain p are not clearly described
5. Table 1 is not necessary, the information is repeated later
6. For more clarity, please enter in tables 2 and 3 the values at T6 and at T12 and the percentages of decrease, not the variation in negative values
7. Please make an analysis of the differences between T6 and T12 in the discussions
8. Please make a stratification of the results according to the two SGLT2i used
Author Response
I congratulate you on this interesting and complex study. I would like you to clarify a few points:
- The title does not reflect the complexity of the studies. Please rephrase
We sincerely thank the reviewer for his appreciation. In accordance with his suggestion, we have reworded the title accordingly with the aim of the study to emphasize the leading result of our investigation, namely the increase in the SMM/FM ratio in patients with type 2 diabetes mellitus treated with SGLT2i.
- In the introduction, the advantages and disadvantages of this class of anti-diabetics should be better pointed out
In accordance with the suggestion, we added a short paragraph describing the main advantages and disadvantages of this important class of drugs. For this reason, we included two more entries (number 7 and number 8 in the current version) and removed two others (number 3 and number 6 in the previous version). The adjustments are highlighted in yellow in the Introduction section.
- Please specify from the patients initially taken into the study, how many remained in the study at T6 and T12?
We thank the reviewer for this comment which allows us to clarify that 2 of the 71 patients (both males) abandoned treatment with SGLT2i (empagliflozin) due to genitourinary complaints. They dropped out after 2 and 3 months of therapy, respectively. Moreover, we could not perform body composition analysis (BIA) in 1 patient at T6 and 2 patients at T12 for instrumental problems. Furthermore, 2 patients at T6 and 3 patients at T12 did not attend the scheduled visits. Missing data were handled with an intention-to-treat approach. All this information has been specified in the paragraph 2.2 of the current version of the manuscript (lines 93-98).
Note that the reduction in the number of patients did not affect the power, which remains above 90%, and that changes over time (T0, T6 and T12) for each variable were analyzed, in the current as in the previous version of the manuscript, with repeated-measures Mixed Models which perform better than other methods in the presence of missing values.
- The statistical methods used to obtain p are not clearly described
Each p-value reported in the results was calculated using the specific test described in the Statistical Analysis section. At the end of this section, we have included the statement that the test is considered significant if p<0.05 (line 185 of the current version of the manuscript)
- Table 1 is not necessary, the information is repeated later
In accordance with the reviewer's suggestion, Table 1 has been removed.
- For more clarity, please enter in tables 2 and 3 the values at T6 and at T12 and the percentages of decrease, not the variation in negative values
Thanks for the suggestion. Tables 2 and 3 have been renumbered as Table 1 and 2, as Table 1 of the previous version of the paper has been removed according to your suggestion. Tables were modified as requested, showing variable means at T6 and T12. The titles and legends have also been changed accordingly. The changes versus T0, expressed as a percentage, are illustrated in Figures 2, 3, 5 and 6
- Please make an analysis of the differences between T6 and T12 in the discussions
The only two variables exhibiting significant changes between T6 and T12 are LDL-cholesterol levels and the FLI value. A comment on these results is included in the discussion from lines 405-410 of the revised version of the manuscript.
.
- Please make a stratification of the results according to the two SGLT2i used
The type of SGLT2i taken by the patients had already been taken into account as a covariate in the statistical analysis and no differences were found in the variation of the variables over time according to the drug used. In the previous version of the article we omitted to describe this step. In the current version, this information has been added both in the statistical analysis section (line 187) and in the results section (lines 271-272).
Round 2
Reviewer 1 Report
Comments and Suggestions for Authors
Figure 4 in the text remains in black and white. Theauthors shuold be revised.
Author Response
Figure 4 in the text remains in black and white. Theauthors shuold be revised.
Thank you for informing us of the inconvenience. To avoid any problems, we have inserted figure 4 in JPEG format into the text. It should now be viewable correctly
Reviewer 2 Report
Comments and Suggestions for Authors
No comments. I agree with the changes
Author Response
No comments. I agree with the changes
Thank you for your valuable suggestions